# Impact of Spatially Heterogeneous Trop-2 Expression on Prognosis in Oral Squamous Cell Carcinoma

**DOI:** 10.3390/ijms23010087

**Published:** 2021-12-22

**Authors:** Ramona Erber, Steffen Spoerl, Andreas Mamilos, Rosemarie Krupar, Arndt Hartmann, Matthias Ruebner, Juergen Taxis, Mareike Wittenberg, Torsten E. Reichert, Gerrit Spanier, Silvia Spoerl

**Affiliations:** 1Institute of Pathology, University Hospital Erlangen, Friedrich-Alexander-Universität Erlangen-Nürnberg (FAU), Comprehensive Cancer Center Erlangen-EMN, 91054 Erlangen, Germany; ramona.erber@uk-erlangen.de (R.E.); arndt.hartmann@uk-erlangen.de (A.H.); 2Department of Cranio-Maxillofacial Surgery, University Hospital Regensburg, 93042 Regensburg, Germany; steffen.spoerl@ukr.de (S.S.); juergen.taxis@ukr.de (J.T.); torsten.reichert@ukr.de (T.E.R.); 3Institute of Pathology, University Hospital Regensburg, 93042 Regensburg, Germany; andreas.mamilos@ukr.de (A.M.); rosemarie.krupar@ukr.de (R.K.); 4Department of Gynecology and Obstetrics, University Hospital Erlangen, Friedrich-Alexander-Universität Erlangen-Nürnberg (FAU), Comprehensive Cancer Center Erlangen-EMN, 91054 Erlangen, Germany; matthias.ruebner@uk-erlangen.de; 5Department of Internal Medicine 5, Hematology/Oncology, University Hospital Erlangen, Friedrich-Alexander-Universität Erlangen-Nürnberg (FAU), Comprehensive Cancer Center Erlangen-EMN, 91054 Erlangen, Germany; mareike.wittenberg@uk-erlangen.de (M.W.); silvia.spoerl@uk-erlangen.de (S.S.)

**Keywords:** Trop-2, TACSTD2, tumor associated calcium signal transducer 2, oral squamous cell carcinoma, prognosis, antibody-drug conjugates, immunohistochemistry, biomarker, head and neck cancer

## Abstract

Oral cancer often presents with aggressive behavior and a high risk of recurrence and metastasis. For oral squamous cell carcinoma (OSCC), which is the most frequent histological subtype, therapy strategies include surgery, radiation therapy, chemotherapy, immune checkpoint inhibitors, and EGFR inhibitors. Recently, a Trop-2 antibody-drug conjugate (ADC) has been approved in the United States of America for the treatment of advanced triple-negative breast cancer. However, this ADC has also been tested in other solid tumors including head & neck squamous cell carcinoma. The prognostic impact of Trop-2 has already been reported for several cancers. We studied the prognostic influence of Trop-2 protein expression on OSCC patients’ survival. The cohort comprised *n* = 229 OSCC patients with available archived tumor tissue and corresponding non-neoplastic oral mucosa tissue. Using immunohistochemistry, we investigated Trop-2 expression in both the central and peripheral regions of each tumor and in corresponding non-neoplastic oral mucosa. In patients suffering from OSCC with combined high central and low peripheral Trop-2 expression, five-year overall survival (OS) was 41.2%, whereas 55.6% of OSCC patients who presented lower central and/or higher peripheral tumoral Trop-2 expression were alive after five years (*p* = 0.075). In multivariate Cox regression, the expression pattern of high central tumoral and lower peripheral Trop-2 expression was significantly correlated with impaired OS (HR = 1.802, 95%-CI: 1.134–2.864; *p* = 0.013) and recurrence-free survival (RFS) (HR = 1.633, 95%-CI: 1.042–2.560; *p* = 0.033), respectively, when adjusting for co-variables. Hence, Trop-2 may serve as an independent prognostic biomarker in OSCC. In subsequent studies, the pathophysiological meaning of downregulated Trop-2 expression in the OSCC periphery has to be analyzed.

## 1. Introduction

In 2020, 377,713 new cases of lip and oral cavity cancer were diagnosed worldwide. Overall, 177,757 people died due to this malignant tumor entity in 2020, However, the incidences differ widely between geographical regions, with high rates in South-Central Asia, parts of Europe, Australia and New Zealand, parts of North and South America, the Caribbean, South Africa and in Pacific regions (e.g., Melanesia) [1,2]. Oral cancer presents with aggressive tumor biology, a high risk of recurrence and metastasis, respectively, and a 5-year survival rate of 50–60%. The most frequent histological subtype (>90%) of oral cancer is oral squamous cell carcinoma (OSCC) (summarized in [3]). According to national and international guidelines, the therapeutic options of OSCC include surgery, radiation therapy and chemotherapy. Topical chemotherapies and immunomodulant treatments have also been proposed but with variable results [4,5]. Furthermore, depending on the expression of the programmed death 1 ligand 1 (PD-L1), a checkpoint inhibitor against programmed death 1 (PD-1; pemrolizumab or nivolumab) or cetuximab, an anti-epithelial growth factor receptor (EGFR) antibody should be considered in the advanced stages of the disease [6,7].

Generally, to improve the therapies and prognosis of cancer patients, continuous investigation of diagnostic, prognostic and predictive biomarkers is indispensable. One interesting biomarker is the human trophoblast cell surface glycoprotein Trop-2. This calcium signal transducer is encoded by the gene *tumor associated calcium signal transducer 2* (*TACSTD2)* [8,9]. Besides expression in trophoblasts, varying expression rates of Trop-2 can be found in non-neoplastic epithelial cells; in stratified squamous epithelium, e.g., tonsil-crypt epithelium, esophagus, cervix and skin; and in cuboidal and columnar epithelial cells, e.g., of the breast, bile ducts, kidney, salivary gland, uterus and prostate [10,11]. Medium Trop-2 expression was furthermore described for the squamous epithelial cells of the oral mucosa [12,13] Moreover, the (over-)expression of Trop-2 had been found in a variety of human malignancies, e.g., cancer of the lung, prostate, pancreatic, gastric, colorectal, breast, cervix, ovarian, esophageal and urinary bladder cancer (summarized in [11,14]). In several cancer entities, Trop-2 protein expression may be used as an independent prognostic marker (summarized in [15]). In OSCC, high Trop-2 protein expression was reported to be associated with tumor grade and poor overall survival [16,17]. One reason may be the role of Trop-2 as an activator of the phosphoinositide 3-kinase (PI3K)/Akt signaling pathway in human OSCC cells [17]. Furthermore, prior studies report that Trop-2 overexpression stimulates cancer cells to proliferate, migrate, invade and metastasize in other solid tumors (e.g., lung cell adenocarcinoma, prostate cancer) [18,19].

Trop-2 has already been tested as a target for precise medicine. Currently, chimeric antigen receptor (CAR) T cell therapy is tested in solid tumors using in vivo models with promising results [20]. Moreover, the antibody drug conjugate (ADC) Sacituzumab Govitecan-hziy, which contains a humanized anti-Trop-2 monoclonal antibody and the topoisomerase I inhibitor drug SN-38 [21,22], was approved by the U.S. Food and Drug Administration (FDA) for the treatment of patients with unresectable locally advanced or metastatic triple-negative breast cancer (mTNBC) [23] based on the results of the randomized phase III ASCENT study. The Trop-2 ADC significantly prolonged both progression-free survival (PFS) and overall survival (OS) in mTNBC patients [24]. In a phase I/phase II study (Study of Sacituzumab Govitecan-hziy (IMMU-132) in Adults With Epithelial Cancer, NCT01631552), the safety, tolerability and efficacy of Sacituzumab Govitecan-hziy has been tested in a variety of epithelial cancers including head & neck squamous cell carcinoma [25,26].

Still, further research is required to analyze the role of Trop-2 in the targeted therapy of OSCC patients. To obtain better insights into the Trop-2 protein distribution within OSCC—as a prerequisite for precise medicine—and into the impact of Trop-2 expression as a prognostic biomarker in OSCC, we evaluated the immunohistochemical (IHC) expression of Trop-2 in the central and peripheral regions of OSCC, Trop-2 expression in the non-neoplastic squamous epithelium of the oral mucosa and its association with the clinicopathological risk factors and survival of OSCC patients.

## 2. Results

### 2.1. Patients

Out of 229 patients in the complete TMA cohort, 160 could be analyzed in a matched-pair approach. The remaining dots did not include invasive tumor, got lost during the staining procedure nor met our criteria of adequate tissue quality. The final data set included 160 patients with clinicopathological characteristics, survival data, as well as Trop-2 IHC expression in all three tissue compartments.

Th patients’ median age was 60.2 years (range: 34.0–90.5 years), and the majority of tumor patients were male (72.5%) (Table 1). 127 patients (79.4%) presented a history of nicotine abuse, whereas 117 patients (73.1%) had a history of alcohol abuse. For the evaluation of comorbidities, the age-adjusted Charlson comorbidity index (ACCI) was evaluated. Hereby, the mean ACCI was 3.0 (range: 0–10). Most tumors were located in the floor of the mouth (46.3%), with the majority of tumors staged UICC class IV (45.6%). Correlation analyses of Trop-2 expression in central and peripheral malignant areas could not reveal any significant correlation for the expression pattern of central high and peripheral low Trop-2 in OSCC patients (Table 1).

### 2.2. Trop-2 Expression in OSCC

Trop-2 protein expression measured by IHC was found to be on the cell surface as membranous staining of the invasive tumor cells and non-neoplastic epithelial cells of the oral mucosa Varying Trop-2 expression of OSCC measured by H-scores are illustrated in Figure 1. Hereby, a median H-score for the central tumor was 226, for the peripheral OSCC 227 and for the non-malignant mucosa 280 (Figure 2). The central and peripheral tumor regions of OSCC had significantly lower H-scores than the corresponding non-neoplastic mucosa (Figure 2 and Figure 3). 34 cases (21.3%) showed a combination of high central Trop-2 expression and low peripheral Trop-2 expression. This phenomenon was noticed in particular when peripheral tumor cells at the invasion border grew more diffusely or in smaller cell clusters compared to a more solid central growth pattern (Figure 4). 8.1% (*n* = 13) of the OSCC presented with combined low central Trop-2 expression and high peripheral Trop-2 expression (Figure 5). In the non-neoplastic mucosa, cells within the stratum basale often showed lower Trop-2 expression compared to the differentiated keratinocytes of the layers above (data not shown).

### 2.3. Overall Survival

The survival analysis revealed a five-year OS of 52.5% in the present OSCC cohort (*n* = 160). The univariate survival analysis for central high and peripheral low Trop-2 expression resulted in a five-year OS of 41.2%, whereas 55.6% of OSCC patients who presented lower central and/or higher peripheral tumoral Trop-2 expression were alive after five years (Figure 6, *p* = 0.075). Additionally, we performed risk-adjustment using multivariate Cox regression. In this regard, patients with an expression pattern of central tumoral high and peripheral low Trop-2 expression displayed a significantly decreased OS (HR = 1.802, 95%-CI: 1.134–2.864; *p* = 0.013). Additionally, elevated age (HR = 1.766, 95%-CI: 1.125–2.773; *p* = 0.013), an ASA-score > 2 (HR = 2.146, 95%-CI: 1.453–3.170; *p* < 0.001) and advanced UICC stages (UICC IV: HR = 2.110, 95%-CI: 1.170–3.804; *p* = 0.013) occurred as relevant determinants of OS in OSCC patients (Table 2).

### 2.4. Recurrence-Free Survival

Regarding disease recurrence, we analyzed RFS in the present TMA cohort of OSCC patients. Univariate survival analysis displayed, analogously to OS, a trend of impaired RFS for patients with high central and low peripheral tumoral Trop-2 expression. Five-year RFS was thus 35.3% for patients with high central and low tumoral Trop-2 expression, whereas cases with lower central and/or higher peripheral tumoral Trop-2 expression displayed a five-year RFS of 50.0% (*p* = 0.093) (Figure 7). In multivariate Cox regression, the expression pattern of high central tumoral and lower peripheral Trop-2 expression was significantly correlated with impaired RFS when adjusting for co-variables (HR = 1.633, 95%-CI: 1.042–2.560; *p* = 0.033) (Table 3).

## 3. Discussion

In the current study, we analyzed the distribution of spatial Trop-2 protein expression in oral squamous cell carcinoma (OSCC) using immunohistochemistry (IHC), correlation with clinicopathological variables and the influence of Trop-2 expression level on OSCC patients’ survival.

Using a primary Trop-2 antibody that binds to the extracellular domain, we found 21.3% of OSCC patients presenting with high central and low peripheral Trop-2 expression. When illuminating this in survival analyses, the expression pattern of high central and low peripheral tumoral Trop-2 expression was associated with poorer patient outcome. Hereby, significantly impaired OS as well as RFS could be seen for this explicit pattern in multivariate survival analysis (Table 2 and Table 3). Other independent prognostic parameters in our OSCC cohort were ASA score and UICC stage.

Still, the diagnosis of malignant oral cancer is associated with relatively high rates of mortality, with age-adjusted death rates estimated at 3–4.1 per 100,000 men and 1.5–2.0 per 100,000 for women in most countries [2]. In 2020, 125,022 men and 52,735 women worldwide died from lip/oral cancer [27]. Hence, the analysis and improvement of diagnostic, prognostic, and predictive biomarkers as well as the development and optimization of therapeutic options, e.g., targeted therapy using ADCs, should be one of the goals in cancer research. ADCs consist of an antibody binding to the target of interest (i.e., an antigen expressed by cancer cells) and a payload, which is the cytotoxic drug (e.g., a tubulin inhibitor). Those components are connected using a linker with a specific binding ability (summarized in [28]). Currently, ADCs are investigated in head & neck cancer (e.g., Trastuzumab Deruxtecan—target: HER2; Losatuxizumab vedotin—target: EGFR) (summarized in [29]). Another ADC that is currently analyzed in head & neck squamous cell carcinoma is Sacituzumab Govitecan-hziy, which targets Trop-2 [26,29]. Future clinical studies have to investigate whether this Trop-2 ADC can add any benefit to the current standard of care. If Sacituzumab Govitecan-hziy shows superiority to other therapies, it has to be additionally analyzed whether Trop-2 is required as predictive biomarker—i.e., as a companion diagnostic in advanced OSCC prior to ADC therapy. In the ASCENT study, which investigated Sacituzumab Govitecan-hziy in mTNBC, patients had been randomized independently from Trop-2 expression. The prespecified biomarker sub-study showed a clinical benefit of Sacituzumab Govitecan-hziy (compared to treatment of physician’s choice) for high or medium Trop-2 expression levels. However, no relevant conclusions could be drawn for low Trop-2 expression due to the small size of the low expression subgroup [30].

Other working groups have already investigated the role of Trop-2 expression in regard to prognosis in OSCC. However, the patient number was partly smaller, and, mostly, they did not compare central vs. peripheral Trop-2 expression. Zhang et al. investigated Trop-2 expression in *n* = 108 cases of OSCC and found an association with decreased overall survival in patients diagnosed with OSCC with high Trop-2 expression. In contrast to our study, they focused on the influence of cytoplasmic expression. To analyze the cytoplasmic expression, they used another primary antibody against Trop-2. Moreover, they did not compare central vs. peripheral expression [16]. Hence, their results are not directly comparable with ours. Tang et al. assessed the expression of Trop-2 in *n* = 187 OSCC. Furthermore, they investigated Trop-2 expression in non-neoplastic oral mucosa, hyperplasia and oral potentially malignant lesions. They defined the Trop-2 H-score cut-off using the X-Tile tool [17,31]. In 62.6% of oral cancers, they observed high Trop-2 expression (H-score cut-off: >120). Normal oral mucosa showed high Trop-2 staining levels in 26.9% [17], which is in contrast to our findings, which indicate elevated Trop-2 expression in oral mucosa when compared to central (*p* < 0.0001) as well as peripheral OSCC (*p* < 0.0001). However, confirming our results, another group reported non-neoplastic stratified epithelial tissue adjacent to squamous cell carcinoma (SCC) from the cervix, esophagus and head and neck, presenting with strong membranous Trop-2 staining of differentiated keratinocytes in the stratum spinosum, stratum granulosum and stratum corneum. Similar to our study, they observed absent Trop-2 staining in less differentiated keratinocytes in the stratum basale. Moreover, their data indicated that Trop-2 loss could be a phenomenon of poorly differentiated SCC at different anatomical sites and that a gradual loss of Trop-2 may be a hallmark of the stepwise progression of SCC [32]. Fong et al. analyzed *n* = 90 OSCC in regard to Trop-2 expression [33]. For the Trop-2 IHC assessment, they determined the immunoreactive score (IRS) according to Remmele and Stegner [34]. Trop-2 overexpression, defined as IRS ≥ 4, was reported in 58.0% of OSCC samples. Overexpression predicted independently poor overall survival [33]. Due to the different population cohorts, primary antibodies and assessment methods used in the studies mentioned above, a direct comparison of their study’s results with our study’s results is not reasonably possible. Further studies might be needed to investigate the meaning of the optimal Trop-2 primary antibody and IHC protocol. Furthermore, it has to be analyzed whether there are differences in regard to Trop-2 expression between Asian and Caucasian OSCC. Several studies on the genomic landscape of Asian and Caucasian OSCC have already revealed distinct molecular differences, which are supposed to be partly due to varying population-specific risk factors (summarized in [35]).

Trop-2 expression, considered as a transmembrane protein, has been described in many non-neoplastic cells but also in cancer cells. Moreover, it has been discussed as a stem/progenitor cell biomarker and reported to be a calcium signal transducer. Trop-2 has a regulatory function within several signaling pathways, including PI3K/Akt, MAPK/ERK, ErbB, TGF-β, Wnt/β-catenin, JAK/STAT, integrin and adherent and tight junction signaling pathways. Due to both the activating and inhibiting capabilities of Trop-2, it is supposed that the function is context-dependent. Next to transmembraneous localization, cytoplasmic expression of Trop-2 can also be found. However, the precise mechanisms determining the cellular localization and its effects on the function of Trop-2 have to be investigated in more detail. Trop-2 has been reported to activate β-catenin and to interact with Claudin 1 and 7 proteins (summarized in [36]).

Still, its role in cancer is controversial (summarized in [36]). In hepatocellular carcinoma, downregulated Trop-2 expression was associated with poorer overall survival [37]. For many other malignant tumors, in contrast, the overexpression of Trop-2 has been found and was associated with poorer patient survival. However, in some cancer entities, e.g., adenocarcinoma of the lung, cervical cancer and head and neck cancer, discrepant results have been found. Whereas some studies reported overexpression, others described downregulated Trop-2 expression (summarized in [36]). Moreover, a correlation with the cellular localization of Trop-2 and its clinical impact has been described [38].

Trop-2 fulfills several functions in cancer that are not fully understood. In some cancer entities, Trop-2 expression has been associated with an epithelial phenotype with a lack of the mesenchymal gene signature but a positive correlation with retained E-cadherin expression. In contrast, Trop-2 expression has also been linked to the mesenchymal phenotype in other cancer entities. Hence, the relationship between Trop-2 and epithelial-mesenchymal transition is contradictory and probably dependent on, among other things, tumor entity (summarized in [36]). Downregulated Trop-2 expression has been reported in squamous cell carcinomas presenting with the molecular and histologic features of EMT. Based on the results from Trop-2 knockout animal experiments, Trop-2 IHC expression was investigated in sarcomatoid head and neck carcinomas. Trop-2 expression was retained in regions with visible squamous cell differentiation but lost in the spindle cell components of sarcomatoid carcinomas [39]. In the present study, we observed this phenomenon of Trop-2 loss in poorly differentiated cancer cells at the invasion border, with tumor cells growing more diffusely or in smaller cell clusters compared to a more solid central growth pattern. Intratumoral Trop-2 heterogeneity and Trop-2 loss in cancer cells with mesenchymal phenotype have also been described for prostate and breast cancer. Remšík et al. cautioned that Trop-2 targeted monotherapy could condition a clonal selection of basically resistant, Trop-2 negative cancer cells that exhibit the mesenchymal phenotype [40]. Moreover, Trop-2 expression may stimulate in some cancers but downregulate tumor cell proliferation in other tumor (sub-)types (summarized in [36]). In OSCC, Trop-2 has been reported to increase tumor growth and metastatic potential through the stimulation of the PI3K/Akt signaling pathway [41]. In our study, we found a combination of high Trop-2 in the tumor center and low Trop-2 expression at the periphery being associated with decreased survival. Hypothetically, one may conclude that Trop-2 promotes both cancer cell proliferation within the central part of the tumor and epithelial-mesenchymal transition at the OSCC invasion border. Another study supports our finding that the peripheral loss of Trop-2 expression has some functional impact on cancer progression. Wang et al. focused on Trop-2 expression in squamous cancer tissues. Membrane localization of Trop-2 expression was associated with stratified epithelial homeostasis; in contrast, Trop-2 loss with poorly differentiated SCC from the cervix, esophagus and head and neck stepwise affects cancer progression and treatment resistance through attenuating chemotherapeutic reagent-induced apoptosis [32]. However, further research on the biological and clinical meaning of intratumorally heterogeneously expressed Trop-2 is needed.

A limitation of the present study is that we analyzed the Trop-2 protein but not mRNA expression. Nevertheless, IHC is an established, time- and cost-effective method that is widely available across different countries and laboratories. Moreover, IHC is often used as method of choice for companion diagnostic tests. Due to the lacking of an invasive tumor, tissue loss during the staining procedure or unmet quality criteria, we could not study the whole TMA cohort but only 69.9% in a matched-pair approach. However, one strength of our study in comparison to two prior studies is that we assessed a larger cohort of OSCC cases. Additionally, we explored the pathophysiological relevance of the intratumoral heterogeneity of Trop-2 expression by showing spatial differences in Trop-2 expression regarding the central vs. peripheral tumor regions.

Several working groups, including us, have shown that Trop-2 may serve as an independent prognostic biomarker in OSCC patients. However, the role of Trop-2 in tumorigenesis and cancer progression needs further investigation, especially when it comes to implementing new therapeutic strategies as specific antibodies.

## 4. Materials and Methods

### 4.1. Patient Selection and Clinical Data

This study involved adult patients examined and treated for a newly diagnosed OSCC at the Department of Cranio-Maxillofacial Surgery, University Hospital Regensburg between the years 2003 and 2014. All participants underwent surgical resection of the primary lesion to negative margins as well as neck dissection based on the clinical and radiological findings. None of the patients received a previous neck dissection or a neoadjuvant treatment for the downstaging of the tumor. All patients were staged according to the UICC (Union internationale contre le cancer) guidelines in the 7th edition [42]. The analysis was done retrospectively, and patient data were retrieved from medical records. The data included age, sex, positive smoking and alcohol anamnesis, tumor site, tumor-node-metastasis (TNM) stage and surgical and adjuvant therapy. Age adjusted Charlson comorbidity index (ACCI) was calculated as previously described and without taking OSCC into account [43]. Adjuvant treatment was based on the recommendation of the multidisciplinary tumor board, and radiotherapy or chemo-radiotherapy was used accordingly. Disease relapse was defined as local disease recurrence or distant metastasis by radiologic evidence with clinical correlation or histologic confirmation by biopsy. Data concerning overall survival (OS) and recurrence-free survival (RFS) were obtained from medical records, death certificates, registration offices and the Clinical Cancer Registry of the Tumor Center—Institute for Quality Management and Health Services Research, University of Regensburg, Germany. The cut-off date was set at 30 June 2020.

### 4.2. Tissue Microarray

A tissue microarray (TMA) was constructed as previously described [44]. The TMA contained the formalin-fixed, paraffin-embedded human OSCC tissues and corresponding non-neoplastic mucosal tissues of 229 adult caucasian patients examined and treated for newly diagnosed OSCC at the Department of Oral and Maxillofacial Surgery, University Hospital Regensburg, Regensburg between 2003 and 2014 [45]. All of the tissue samples were retrieved from the archive of the Institute of Pathology, University of Regensburg, Germany. Two experienced head and neck pathologists reviewed all hematoxylin and eosin (H&E) sections from all of the candidate specimens to confirm the histological tumor type and grade in each case and select the optimal slides. To map the heterogeneity of the tumor and the tumor microenvironment from each patient, three cores were included: the tumor center, the peripheral invasion front and the adjacent normal tissue. Morphologically representative areas suited for subsequent punching were annotated on H&E stained slides. From the corresponding tissue blocks, cores for the TMA were harvested by using a punch biopsy with a diameter of 2 mm. These cores were taken out of the donor blocks and placed coordinately into recipient paraffin microarray blocks.

### 4.3. Trop-2 Immunohistochemistry

For Trop-2 staining, the IHC protocol was performed on a Ventana Benchmark Ultra automated platform (Ventana Medical Systems, Inc., Oro Valley, AZ, USA). 2 μm thick sections of the TMA FFPE blocks were mounted on adhesive glass slides. We used a primary human monoclonal antibody against Trop-2 (clone: 01; host: mouse; isotype: IgG1; ENZ-ABS380; dilution 1:500; Enzo Life Sciences AG, Lausen, Switzerland) which was applied for 32 min at 37 °C after pretreatment for 8 min with Protease 1 (Roche Diagnostics Schweiz AG, Rotkreuz ZG, Switzerland). The binding of the antibody to the Trop-2 antigen was visualized using the optiView DAB IHC Detection Kit (Ventana) and, subsequently, sections were counterstained with hematoxylin and Bluing Reagent (Ventana). For the establishment of the IHC protocol, human colorectal cancer, human non-neoplastic urothelium, human non-neoplastic oral mucosa and FFPE cell pellets of Trop-2 positive cell lines [the human breast cancer cell lines MCF7 (HTB-22) and SK-BR-3 (HTB-30), and the human choriocarcinoma cell line BeWo (each ATCC, Wesel, Germany)] were used as positive controls [10,12,46,47,48,49,50].

After the Trop-2 staining procedure, the TMA slides were evaluated with a Zeiss Axio microscope (magnification of ×50, ×100, and ×200) by a board-certified pathologist (RE) blinded to clinicopathological and outcome data. First, H&E slides, stained according to the standard in-house protocol, were reviewed in regard to the morphology and growth pattern of OSCC. For each case, membranous staining of the cells of interest (separately for central vs. peripheral cancer cells vs. non-neoplastic epithelial cells) was assessed. To receive the Trop-2 IHC H-score, the percentage (0–100%) for each staining intensity (no staining = 0; weak = 1+, moderate = 2+, and strong = 3+) of stained tumor cells and non-neoplastic cells, respectively, was assessed semi-quantitatively. H-score (possible range from 0–300) was calculated as follows: percentage of cell without staining (0) × 0 + percentage of cells with weak staining intensity (1+) × 1 + percentage of cells with moderate staining intensity (2+) × 2 + percentage of cells with strong staining intensity (3+) × 3 [51,52]. For each region of interest (tumor center vs. tumor periphery vs. non-neoplastic mucosa), H-score values were further median-split and classified into a low (≤median) and a high (>median) Trop-2 expression subgroup for subsequent analyses.

### 4.4. Statistics

The primary objective was to analyze whether the tissue-based biomarker Trop-2, assessed as IHC expression using the H-score, had prognostic value on OS and RFS in addition to well-known prognostic patient and tumor characteristics. OS was defined as the time from the date of primary diagnosis to the earliest date of death from any cause or the date of censoring. Patients who were lost to follow-up before the maximal observation time of 10 years or were alive after the maximal observation time were censored at the last date they were known to be alive or at the maximum observation time. RFS was defined in a similar fashion including the events of distant metastasis and local recurrence.

A multivariable Cox regression model was fitted with OS and RFS separately as outcome and the following predictors: age at diagnosis (continuous), tumor stage (ordinal, T1 to T4), cervical lymph node status (categorical; N0, N+) and Trop-2 H-score class. The proportional hazards assumptions were checked using the method of Grambsch and Therneau [53]. Survival rates were estimated using the Kaplan–Meier product limit method.

When the Trop-2 IHC H-score class was significant in the multivariable Cox model, hazard ratios (HR) were calculated for patient subgroups defined by Trop-2 using the interaction model. In case of non-significance, interactions were removed from the model and HRs from this reduced model were extracted for Trop-2.

All tests were two-sided, and *p <* 0.05 was regarded as statistically significant. All analyses were performed using IBM SPSS Statistics Version 26.0 (IBM Corp., Armonk, NY, USA).

## 5. Conclusions

In our study, OSCC with combined high central and low peripheral Trop-2 expression were significantly associated with poorer overall survival and recurrence-free survival. Hence, Trop-2 may serve as an independent prognostic biomarker in OSCC. Further studies have to analyze the pathophysiological meaning of downregulated Trop-2 expression in the OSCC periphery.

## Figures and Tables

**Figure 1 ijms-23-00087-f001:**
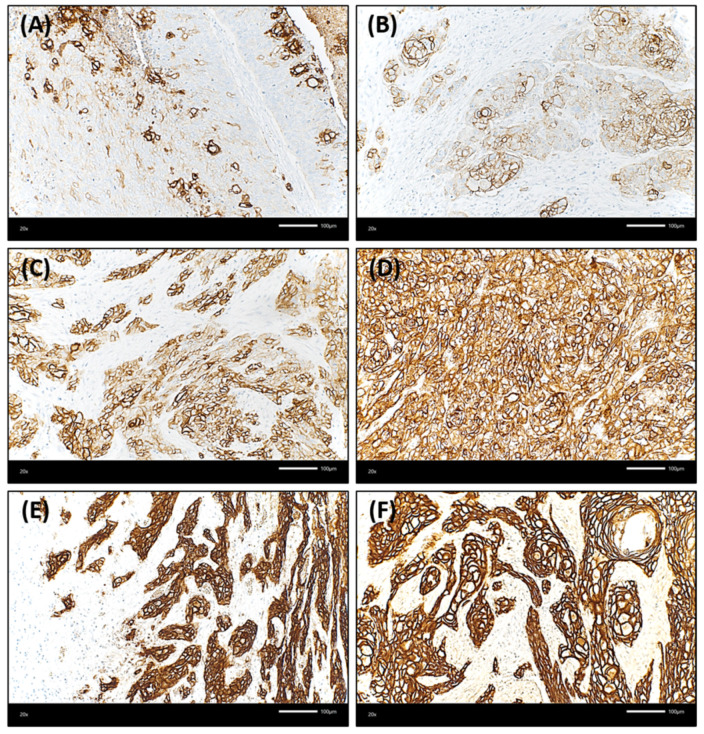
Membranous staining of Trop-2 in oral squamous cell carcinoma (OSCC) using immunohistochemistry (Trop-2 immunohistochemistry, ×200 magnification). OSCC cases presented with membranous Trop-2 expression; however, the intensity and percentage of positively stained cancer cells was variable. (**A**,**B**): Few OSCC samples showed with overall low but heterogenous Trop-2 expression, with multiple cancer cells presenting with a loss of or only low Trop-2 staining admixed with tumor cells with strong Trop-2 staining intensity (H-score of 75 and 100, respectively). (**C**): OSCC with heterogenous but medium Trop-2 expression (H-score = 175). Single cells present with very low staining. (**D**) Homogenous, medium Trop-2 expression of an OSCC case (H-score = 190). (**E**,**F**): Two OSCC with high Trop-2 expression (H-scores ≥ 240). The tumor cells show a strong and membranous Trop-2 staining intensity.

**Figure 2 ijms-23-00087-f002:**
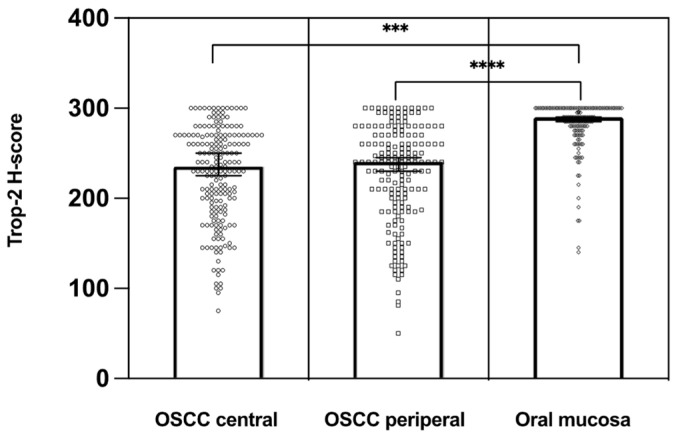
Distribution of Trop-2 H-scores in the central and peripheral tumor regions of OSCC compared to the corresponding non-neoplastic epithelial cells of the oral mucosa. Trop-2 expression was significantly higher in the normal oral mucosa than on cancer cells (*n* = 160). *** *p* < 0.01, **** *p* < 0.001.

**Figure 3 ijms-23-00087-f003:**
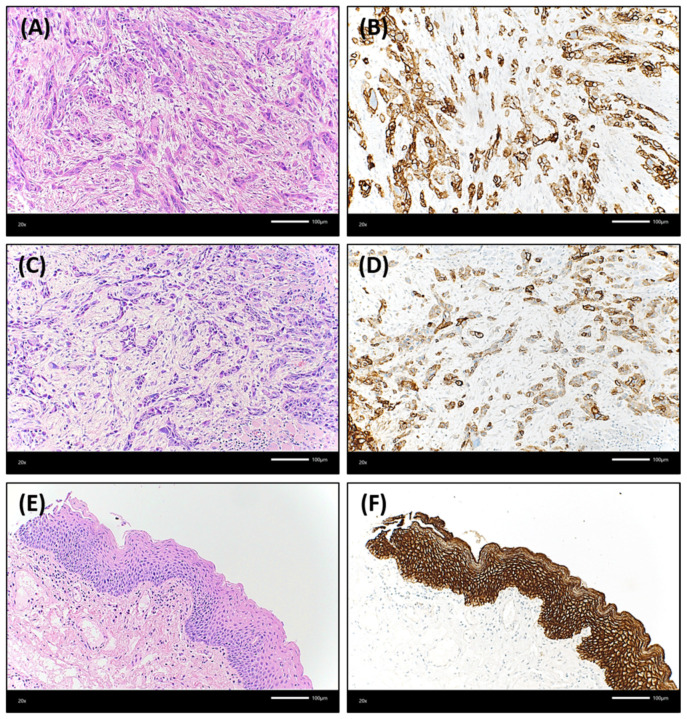
Illustration of an OSCC case that showed a lower Trop-2 expression in both the central and peripheral tumor regions compared to non-neoplastic oral mucosa. Both (**A**) central and (**C**) peripheral tumor cells grew in relatively thin strands or small cell clusters with marked desmoplastic stromal reaction between the epithelial differentiated cancer cells [hematoxylin & eosin (H&E), each ×200 magnification]. (**B**): The central tumor proportion presented with a Trop-2 H-score of 225. 25% of the tumor cells showed a low expression in this area, 25% a medium staining intensity (Trop-2 immunohistochemistry, ×200 magnification). (**D**): In the peripheral region of this OSCC, 20% of tumor cells were without any Trop-2 staining, 65% presented with low or medium staining intensity resulting in a Trop-2 H-score of 115 (Trop-2 immunohistochemistry, ×200 magnification). In contrast, the non-neoplastic oral mucosa (**E**) (H&E, ×200 magnification) had a strong and homogenous Trop-2 expression (**F**) (H-score of 300; Trop-2 immunohistochemistry, ×200 magnification).

**Figure 4 ijms-23-00087-f004:**
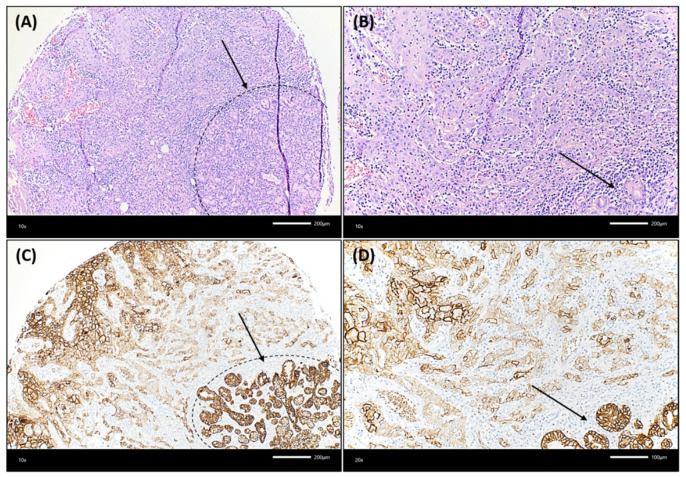
Illustration of an OSCC case that showed lower Trop-2 expression in the very peripheral tumor cells compared to more central cancer cells. (**A**,**B**): The squamous cell carcinoma showed a rather solid growth pattern centrally, whereas tumor cells grew in smaller cell clusters at the very peripheral invasion border, which was located near non-neoplastic salivary glands. The dashed black line in (**A**) and the black arrow in (**A**,**B**) highlight the seromucous salivary glands (hematoxylin & eosin, each ×200 magnification). In (**C**,**D**), a decrease of Trop-2 expression on cancer cells growing in smaller cell clusters at the invasion border, close to salivary glands, can be detected. This is in contrast to the more centrally located larger tumor cell nests, which show a higher Trop-2 expression. The dashed black line in (**C**) and the black arrow in (**C**,**D**) highlight the seromucous salivary glands (Trop-2 immunohistochemistry, each ×200 magnification).

**Figure 5 ijms-23-00087-f005:**
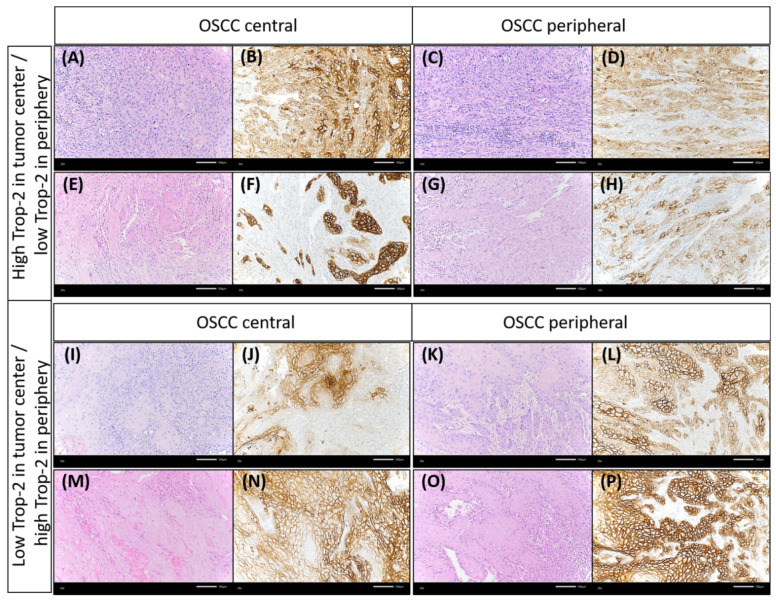
Discrepancy of Trop-2 expression between different tumor regions (central vs. periphery). Some OSCC cases presented with higher Trop-2 expression in the tumor center than in the periphery. Case 1: (**A**,**B**) representing the tumor center, (**C**,**D**) showing the peripheral tumor region. Case 2: (**E**,**F**): tumor center, (**G**,**H**): peripheral tumor region. Few cases showed lower Trop-2 expression in the tumor center than in the periphery. Case 3: (**I**,**J**) representing the tumor center, (**K**,**L**) showing the peripheral tumor region. Case 4: (**M**,**N**): tumor center, (**O**,**P**): peripheral tumor region [(**A**,**C**,**E**,**G**,**I**,**K**,**M**,**O**): hematoxylin & eosin; the remainder: Trop-2 immunohistochemistry, each ×200 magnification].

**Figure 6 ijms-23-00087-f006:**
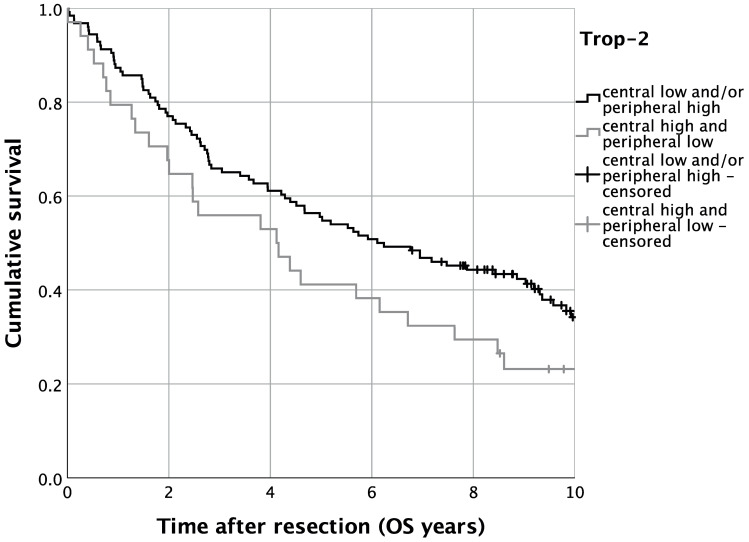
Kaplan–Meier analysis for varying Trop-2 H-scores for overall survival (OS). *n* = 160. *p* = 0.075.

**Figure 7 ijms-23-00087-f007:**
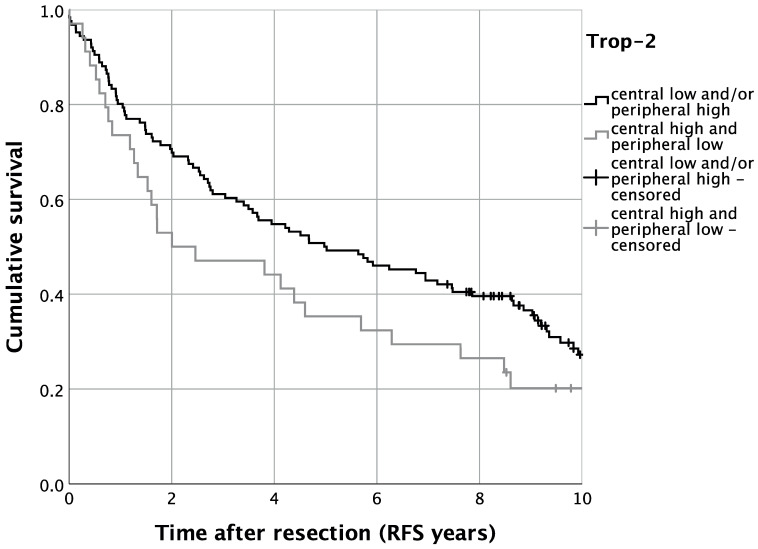
Kaplan–Meier analysis for varying Trop-2 H-scores for locoregional recurrence-free survival (RFS). *n* = 160. *p* = 0.093.

**Table 1 ijms-23-00087-t001:** Patients’ clinicopathological characteristics according to Trop-2 expression in the central and peripheral tumor in the analyzed TMA cohort of *n* = 160 OSCC patients, UICC 7th edition.

		Trop-2 IHC Expression				
		Central Low and/or Peripheral High		Central High and Peripheral Low		χ^2^
		N	%	N	%	*p*
Sex	Female	37	84.1%	7	15.9%	0.309
	Male	89	76.7%	27	23.3%	
Age at diagnosis	<50	20	80.0%	5	20.0%	0.773
	50.0–59.9	44	81.5%	10	18.5%	
	60.0–69.9	31	72.1%	12	27.9%	
	70.0–79.9	24	80.0%	6	20.0%	
	80.0+	7	87.5%	1	12.5%	
Positive anamnesis smoking	No	26	78.8%	7	21.2%	0.995
	Yes	100	78.7%	27	21.3%	
Positive anamnesis alcohol	No	35	81.4%	8	18.6%	0.620
	Yes	91	77.8%	26	22.2%	
Age-adjusted Charlson Comorbidity index	0	10	83.3%	2	16.7%	0.714
	1	25	75.8%	8	24.2%	
	2	25	89.3%	3	10.7%	
	3	25	73.5%	9	26.5%	
	4	15	75.0%	5	25.0%	
	5+	26	78.8%	7	21.2%	
ASA class	I	7	77.8%	2	22.2%	0.549
	II	54	75.0%	18	25.0%	
	III	65	82.3%	14	17.7%	
	IV	0	0.0%	0	0.0%	
Anatomical site	Buccal mucosa	14	82.4%	3	17.6%	0.759
	Upper alveolus and gingiva	5	71.4%	2	28.6%	
	Lower alveolus and gingiva	26	70.3%	11	29.7%	
	Hard palate	6	85.7%	1	14.3%	
	Tongue	15	83.3%	3	16.7%	
	Floor of mouth	60	81.1%	14	18.9%	
Tumor size	T1	34	82.9%	7	17.1%	0.194
	T2	54	84.4%	10	15.6%	
	T3	7	70.0%	3	30.0%	
	T4	31	68.9%	14	31.1%	
Cervical lymph node metastasis	N0	66	81.5%	15	18.5%	0.442
	N1	26	81.3%	6	18.8%	
	N2/3	34	72.3%	13	27.7%	
Grading	G1	6	100.0%	0	0.0%	0.361
	G2	105	77.2%	31	22.8%	
	G3/4	15	83.3%	3	16.7%	
UICC stage	I	20	74.1%	7	25.9%	0.150
	II	28	87.5%	4	12.5%	
	III	25	89.3%	3	10.7%	
	IV	53	72.6%	20	27.4%	
Adjuvant therapy	No	54	79.4%	14	20.6%	0.068
	Radiotherapy	55	84.6%	10	15.4%	
	Radiochemotherapy	17	63.0%	10	37.0%	
Death	Alive	39	84.8%	7	15.2%	0.236
	Dead	87	76.3%	27	23.7%	
Death or recurrence	Alive without recurrence	33	84.6%	6	15.4%	0.303
	Death or recurrence	93	76.9%	28	23.1%	

Abbreviations: ASA: American Society of Anesthesiologists; UICC: Union internationale contre le cancer.

**Table 2 ijms-23-00087-t002:** Multivariate Cox regression for Trop-2 central high and peripheral low as well as prognostic factors on OS (*n* = 160). UICC 7th edition.

Variable	Category	*p*	HR	Lower 95%-CI	Upper 95%-CI
Sex	Female		1.000		
	Male	0.266	1.335	0.802	2.220
Age at diagnosis	≤70		1.000		
	>70	0.013	1.766	1.125	2.773
ASA class	≤2		1.000		
	>2	<0.001	2.146	1.453	3.170
Alcohol anamnesis	No		1.000		
	Yes	0.647	0.891	0.543	1.461
Smoking anamnesis	No		1.000		
	Yes	0.431	0.789	0.438	1.423
UICC stage	I		1.000		
	II	0.163	1.630	0.820	3.237
	III	0.127	1.727	0.856	3.487
	IV	0.013	2.110	1.170	3.804
Grading	G1		1.000		
	G2	0.932	0.954	0.324	2.808
	G3/4	0.877	1.101	0.327	3.707
Trop-2 expression profile	Central low and/or peripheral high		1.000		
	Central high and peripheral low	0.013	1.802	1.134	2.864

Abbreviations: HR: hazard ratio; CI: confidence interval; OS: overall survival; ASA: American Society of Anesthesiologists; UICC: Union internationale contre le cancer.

**Table 3 ijms-23-00087-t003:** Multivariate Cox regression for Trop-2 central high and peripheral low as well as prognostic factors on RFS, (*n* = 160), UICC 7th edition.

Variable	Category	*p*	HR	Lower 95%-CI	Upper 95%-CI
Gender	Female		1.000		
	Male	0.268	1.329	0.803	2.201
Age at diagnosis	≤70		1.000		
	>70	0.089	1.475	0.942	2.310
ASA class	≤2		1.000		
	>2	<0.001	2.016	1.385	2.934
Alcohol anamnesis	No		1.000		
	Yes	0.669	0.899	0.553	1.463
Smoking anamnesis	No		1.000		
	Yes	0.130	0.640	0.359	1.140
UICC stage	I		1.000		
	II	0.492	1.264	0.648	2.465
	III	0.140	1.664	0.846	3.272
	IV	0.035	1.839	1.045	3.235
Grading	G1		1.000		
	G2	0.798	1.150	0.394	3.361
	G3/4	0.531	1.470	0.441	4.9001
Trop-2 expression profile	Central low and/or peripheral high		1.000		
	Central high and peripheral low	0.033	1.633	1.042	2.560

Abbreviations: CI: confidence interval; HR: hazard ratio; RFS: recurrence-free-survival; ASA: American Society of Anesthesiologists; UICC: Union internationale contre le cancer.

## Data Availability

Data can be obtained by scientists that work independently from the industry on request. Data are not stored on publicly available servers.

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
