# Peer review of "Impact of Spatially Heterogeneous Trop-2 Expression on Prognosis in Oral Squamous Cell Carcinoma"

_ijms, 2021, doi:10.3390/ijms23010087_

Round 1
Reviewer 1 Report
An interesting original article studying the prognostic influence of Trop-2 protein expression on OSCC patients’ survival, indicating poorer overall survival and recurrence-free survival, and suggesting that Trop-2 may serve as an independent prognostic biomarker in OSCC patients.
Although various other similar studies are present in literature, given the good number of participants this article may be eligible to be published after minor revisions:
line 259 "Still, diagnosis of malignant oral cancer is associated with relatively high rates of mortality " please specify the rates reported.
Page 2 line 54 you should add: " topical chemotherapies and immunomodulant treatments have also been proposed, but with variable results" and cite an article such as: doi: 10.3390/medicina57060563. and doi: 10.3390/curroncol28040213.
Thank You
Reviewer 2 Report
Dear authors,
thank you for the opportunity to review this interesting study on the significance of Trop-2 in oral squamous cell carcinoma. There is an urgent need for clinically relevant biomarkers in OSCC and the presented study provides interesting data on Trop-2 by analyzing a large cohort of surgically treated tumors. The tables and figures are clear and informative, the manuscript is written appealingly and the data seems sound.
Some minor points that may be considered:
I would suggest including some numerical data in the abstract to allow for a quick scan of the results (e.g. concrete results of survival analysis OS and RFS).
Table 1 shows the distribution of the Trop-2 groups in dependence of several parameters including the adjuvant therapy. As I understand it, original tissue from the primary tumors were used for this analysis, I do not see the significance of correlating Trop-2 data from this tissue with the extent of adjuvant therapy - how should there be a relevant connection (especially if correlations with T- or N-stage is missing)?
As it comes to the survival analysis as I am familiar with it, only those factors should be included in the multivariate analysis, that showed a significant impact in the univariate survival analysis. How did you conduct this analysis, as e.g. in case of RFS, you included ASA class, age etc. - did you see any univariate impact here?
Please clarify how many patients were included in each group (the Trop-2 distribution) for each analysis. In the results section I see one group with 34 patients and one group with 13 patients. Does that mean that out of 229/160 patients, only 47 were included in the survival analysis (in table 1 160 patients were included)? Maybe a short table could help to clarify the group distributions.
